# Sequence-based generative AI design of versatile tryptophan synthases

Théophile Lambert [1,5], Amin Tavakoli[2], Gautham Dharuman[3], Jason Yang [1], Vignesh Bhethanabotla [1], Sukhvinder Kaur[4], Matthew Hill[4], Arvind Ramanathan [3], Anima Anandkumar [2] ✉ & Frances H. Arnold [1] ✉

Enzymes are powerful and sustainable catalysts, but their widespread application is limited by the difficulty of identifying functional starting points for optimization, creating a major bottleneck in early-stage biocatalyst discovery. Designing libraries of such starting enzymes remains particularly challenging. Here, we use the GenSLM protein language model to generate novel $\beta$-subunit of tryptophan synthase (TrpB) enzymes that express in *Escherichia coli* and are both stable and catalytically active. Many generated TrpBs also display significant substrate promiscuity, outperforming their natural counterparts on non-native substrates. Some even surpass laboratory-evolved TrpBs. Comparison of the most-active and most-promiscuous generated TrpB to its closest natural homolog confirms that the enhanced versatility is absent from the natural enzyme, highlighting the creative potential of generative models. These results demonstrate that the generated TrpBs not only preserve natural structure and function but also acquire non-natural properties, establishing generative models as powerful tools for biocatalyst discovery and engineering.

Enzymes are exceptionally powerful, selective, and versatile catalysts for efficient and sustainable production of chemicals, fuels, materials, and pharmaceuticals, offering attractive alternatives to traditional chemical methods[1–4]. However, to meet the performance requirements of industrial applications, enzymes often require tailored optimization. In this context, directed evolution (DE) has emerged as a robust and broadly applicable strategy, using iterative rounds of mutagenesis and screening to progressively improve an enzyme toward a desired function[5]. Its success in producing industrially viable biocatalysts has been well documented, and recent advances in AI-guided design and laboratory automation can further streamline the DE workflow, significantly accelerating the pace and efficiency of enzyme engineering[6–9].

Despite its broad applicability, directed evolution (DE) remains fundamentally limited by the need for a starting enzyme with measurable activity toward the target function. Identifying an enzyme with such initial activity is still largely empirical, with no universal strategy currently available. One common approach is to repurpose an existing enzyme by exploiting its ability to catalyze reactions or accept substrates beyond its native biological context, a property known as catalytic or substrate promiscuity[10]. This typically begins by hypothesizing enzyme families that could accommodate the desired transformation based on mechanistic or substrate similarities. Libraries of enzymes are then constructed either by sampling natural sequence diversity or by creating mutant libraries from a small subset of representative enzymes. Both sampling natural sequences and mutating a given enzyme present challenges: natural enzymes may express poorly or have narrow specificity, while mutagenesis covers only a restricted sequence space and frequently yields a high proportion of inactive variants. The process is labor-intensive and time-consuming, with outcomes largely dictated by chance and the composition of the available enzyme libraries. Although DE has helped

[1]Division of Chemistry and Chemical Engineering, California Institute of Technology, Pasadena, CA, USA. [2]Department of Computing and Mathematical Sciences, California Institute of Technology, Pasadena, CA, USA. [3]Argonne National Laboratory, Lemont, IL, USA. [4]Elegen Corp, San Carlos, CA, USA. [5]Present address: Institut de Chimie Moléculaire et des Matériaux d'Orsay (ICMMO), Université Paris-Saclay, CNRS UMR8182, Orsay, France. ✉e-mail: anima@caltech.edu; frances@cheme.caltech.edu

unlock many useful biocatalysts[11,12], many promising transformations remain unexplored, and the uncertainty and duration of this process continue to limit the broader industrial adoption of enzymes as catalysts, particularly when compared to the speed and reliability of conventional synthetic chemistry[2,4].

To address the challenge of identifying novel enzymes with desired functions, we propose to use protein language models (PLMs) to generate libraries of enzymes that can be screened for target activities[9,13,14]. PLM-generated proteins offer significant advantages over natural ones: they can explore regions of sequence space far from known proteins, while also allowing conditioning or filtering to incorporate desirable features[15–18]. Recent experimental applications have validated this approach: PLMs can generate functional proteins with real-world relevance, marking a turning point for machine learning in protein engineering[13,19–22]. Building on this promise, we used the GenSLM model, originally developed for genome-scale applications. Unlike most protein language models, which are trained on amino acid sequences, GenSLM learns interactions within DNA sequences at the codon level. However, experimental validation of DNA-sequence-based models has so far been limited and remains an open area for exploration[15,22,23].

To assess the model's generative potential, we selected a mechanistically challenging enzyme as a test case. We focused on the β-subunit of tryptophan synthase (TrpB), a subunit of the hetero-tetrameric tryptophan synthase complex (TrpS), a longstanding model in mechanistic enzymology. TrpS consists of two TrpA and two TrpB subunits that together catalyze a multistep biosynthetic transformation involving at least nine distinct chemical steps. TrpA produces indole, which is channeled to TrpB through a 20–25 Å substrate tunnel, where it reacts with L-serine to form L-tryptophan. This latter transformation requires the pyridoxal phosphate (PLP) cofactor and a finely tuned network of catalytic residues. Efficient catalysis depends on large-scale conformational dynamics between and within subunits, which regulate transitions between low-activity (open) and high-activity (closed) states, enabling the control of substrate binding, intermediate stabilization, and product release. The TrpB catalytic cycle is shown in Fig. 1B[24].

TrpB is an attractive industrial biocatalyst that provides an efficient and scalable route for synthesizing noncanonical amino acids[25]. A major breakthrough for its usage was the creation of *Pf*TrpB-0B2, a stand-alone catalyst whose activity is no longer dependent on its natural partner TrpA[26]. This greatly enhanced its evolvability and opened

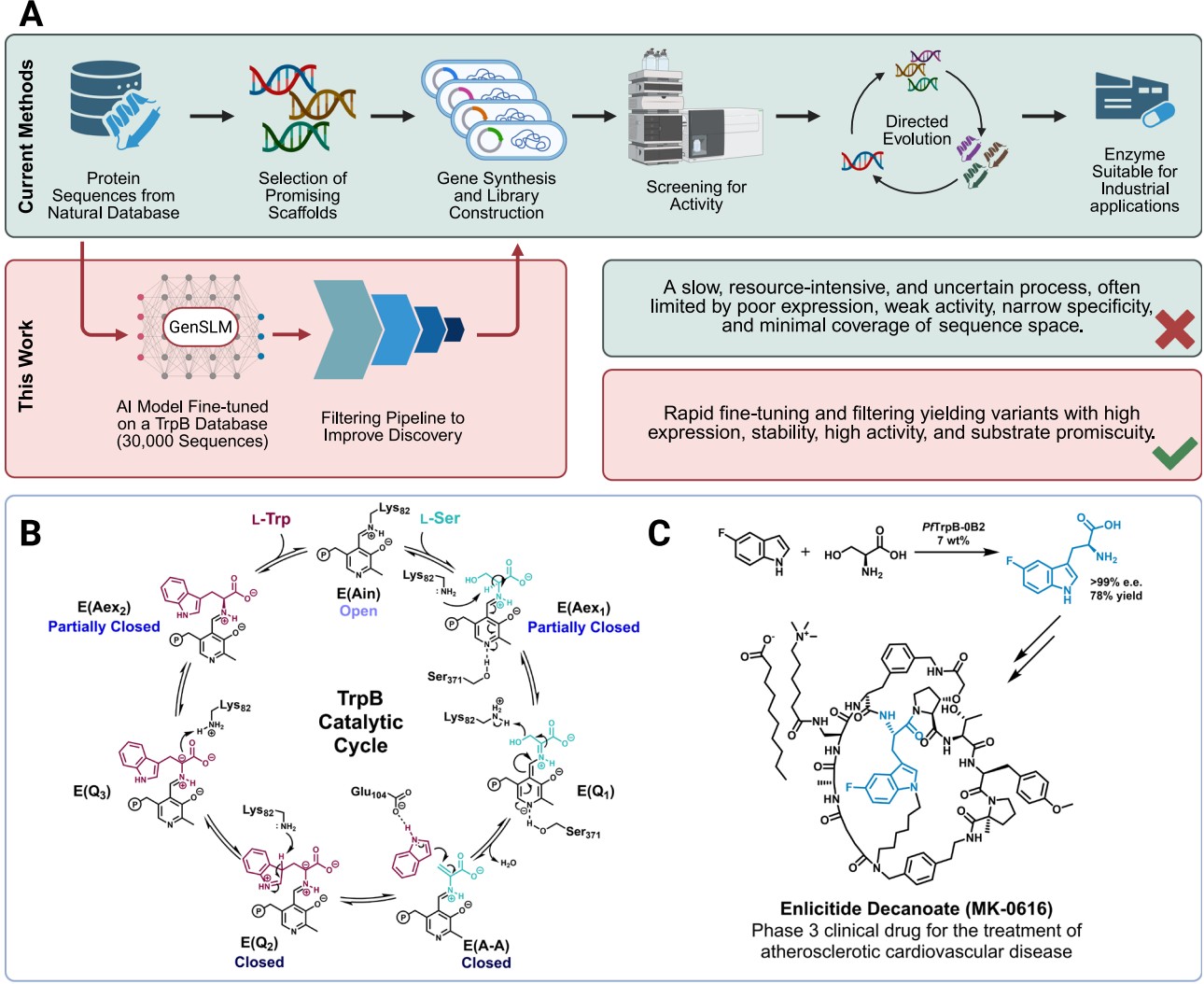

**Fig. 1 | GenSLM-TrpBs: a workflow from in silico design to industrially relevant catalysts. A** Conventional strategies for identifying starting points for directed evolution (DE) typically rely on the intrinsic promiscuity of natural enzymes and require substantial time and experimental efforts. In contrast, libraries generated by GenSLM exhibit favorable properties, including high expression, stability, and catalytic activity, as well as broadened substrate promiscuity, offering an efficient alternative for rapid biocatalyst discovery. **B** Catalytic cycle of TrpB (residue numbering based on *Pf*TrpB). **C** An example of industrial application of engineered TrpBs in the total synthesis of enlicitide decanoate[39]. Created in BioRender, https://BioRender.com/g3hjmd8.

the door to directed evolution efforts aimed at expanding its catalytic scope. Subsequent engineering campaigns have successfully broadened the substrate range of TrpB to include indole derivatives, indole isosteres, non-indolic aromatic compounds, aliphatic substrates, and L-serine derivatives[27–30]. In parallel, in vivo evolution using the OrthoRep system yielded active TrpB orthologs with a variety of distinct promiscuity profiles[31]. Engineered TrpBs have also been coupled with other enzymes to enable the biosynthesis of valuable molecules[32]. Computational approaches, such as enhanced-sampling molecular dynamics and shortest path map analysis, have helped rationalize these experimental findings by identifying key residues and conformational ensembles associated with stand-alone activity[33–35]. Ancestral Sequence Reconstruction (ASR) has also been leveraged to investigate the evolutionary trajectory and allosteric regulation of TrpA-TrpB complexes[36,37]. These studies revealed that ancestral TrpBs were inherently stand-alone enzymes, a property that has since been exploited to engineer solvent-resistant TrpBs[38]. These engineered TrpBs are now used at industrial scale: AralezBio (San Leandro, CA USA) manufactures tryptophan analogs using evolved TrpBs, and Merck & Co. (Rahway, NJ USA) employs *Pf*TrpB-0B2 in the synthesis of 5-fluorotryptophan, a key building block in the production of enlicitide decanoate, a phase 3 clinical candidate (Fig. 1C)[39].

In this work, we show that GenSLM can generate TrpB homologs that are stable, expressible in *E. coli*, catalytically active, and substrate-promiscuous, outperforming natural TrpBs and, in several cases, even engineered variants. These findings demonstrate that PLM-generated homologs can diverge from the natural sequences in the training data by acquiring beneficial properties such as high promiscuity and activity. Generative models provide a powerful alternative to activity-based screening of natural enzyme for discovery of new activities and have the potential to substantially accelerate biocatalyst discovery and development.

## Results

### Generation of TrpBs using GenSLM

The generative AI framework used here is built on the Genome-Scale Language Model (GenSLM), a large-scale transformer architecture designed to capture biological sequence patterns at the codon level. Unlike most PLMs that operate on amino acid tokens, GenSLM represents sequences as contiguous triplets of nucleotides (64 codons), directly mirroring the translation process from DNA to protein. This codon-level representation not only aligns with the central tenet of molecular biology but also enables the model to incorporate synonymous substitutions while learning protein-level effects. GenSLM was trained at multiple parameter scales, 25 million (25M), 250 million (250M), 2.5 billion (2.5B), and 25 billion (25B), using a dataset of approximately 110 million prokaryotic gene sequences from the Bacterial and Viral Bioinformatics Resource Center (BV-BRC)[40]. One application of this model was to study the evolutionary dynamics of SARS-CoV-2, where fine-tuning on 1.5 million viral genomes allowed it to predict variant fitness, anticipate emerging lineages, and identify functionally relevant mutations[23].

To generate TrpB sequences, we employed the 25M parameter GenSLM and fine-tuned it on a curated dataset of *trpB* DNA sequences obtained from BV-BRC[40] containing 30,000 unique *trpB* nucleotide sequences corresponding to 22,800 unique amino acid sequences after translation. Fine-tuning followed the same procedure developed for SARS-CoV-2, using a contrastive learning objective similar to the masked language modeling strategy implemented in SpanBERT with parameters given in the **Methods**[41].

Building on established sequence- and structure-based criteria[42], the generated proteins were subjected to a minimal set of filters designed to remove unpromising sequences. Although more physics informed filters (including docking and thermostability filtering) were initially explored, they were ultimately not employed to avoid

introducing biases that might not correlate with experimental outcomes. Instead, the filtering pipeline focused on evaluating structural and sequence integrity, and promoting diversity and novelty across sequence space, thereby enriching the pool of candidates for subsequent experimental validation. We assembled a reference database of >57,000 natural TrpB sequences, including the 22,800 sequences used for fine-tuning and additional UniProt entries retrieved by querying the gene name "trpb" and restricting the sequence length to 200–600 amino acids. This dataset provided both a baseline for integrity comparisons and a diversity control for filtering. Generated sequences were first filtered by length for consistency with natural TrpBs, then modeled using ESMFold[43]; only structures with a predicted Local Distance Difference Test (pLDDT) score above 0.8 were retained. Each sequence's maximum sequence identity (MaxID) to the reference set was computed and binned as [100–90%], [90–80%], [80–70%], [70–60%], [60–50%], and [50–40%]. To prioritize divergent and potentially novel candidates, sequences with MaxID above 90% were excluded. Conservation of the catalytic lysine, covalently bound to PLP in the resting state and essential for catalysis, was assessed. Prior to final selection, we ensured that candidate sequences originated from distinct BLAST similarity clusters to avoid redundancy and to maximize coverage of diverse regions in sequence space.

From this filtered set, we selected 105 representative sequences distributed as follows: 30 sequences with 80–90% MaxID, 40 with 70–80%, 20 with 60–70%, 10 with 50–60%, and 5 with 40–50%. This distribution was intentionally biased toward higher-identity sequences, which are generally associated with a greater likelihood of activity, while still preserving lower-identity sequences to explore broader sequence diversity. The final set of *E. coli* optimized DNA sequences, designated GenSLM-TrpBs in this article, was synthesized by Elegen Corp (Menlo Park, CA USA) and all sequences are provided in Supplementary Data 1.

Embeddings from the GenSLM model were used to compare the distribution of generated sequences with that of natural TrpB sequences. The generated sequences broadly spanned the natural sequence space used for fine-tuning, as shown by the t-distributed stochastic neighbor embedding (t-SNE) plot (Fig. 2A). Principal component analysis (PCA) and uniform manifold approximation and projection (UMAP) analyses of the same dataset are also provided (Supplementary Fig. S2). Positional variability was highly similar between natural and generated sequences, and key conserved residues in natural TrpBs were retained in the generated sequences (Fig. 2B, C). Together, these results indicate that the model successfully captures the structural and evolutionary constraints that define the TrpB sequence landscape.

### GenSLM-TrpBs catalyse tryptophan formation with high stability and robust expression

Our first objective was to evaluate whether the GenSLM-TrpBs exhibited activity for tryptophan synthesis. To this end, we expressed the 105 selected GenSLM-TrpBs in *Escherichia coli* and compared their performance to several well-characterized natural and engineered TrpB enzymes. Specifically, we selected natural TrpBs from diverse organisms available in our laboratory culture collection, which are known for their reliable heterologous expression: *Escherichia coli* (*Ec*TrpB), *Arabidopsis thaliana* (*At*TrpB), *Pyrococcus furiosus* (*Pf*TrpB), *Thermotoga maritima* (*Tm*TrpB), and *Streptomyces albus* (*Sa*TrpB). *Pf*TrpB-0B2 was included as a laboratory-evolved and stand-alone TrpB[26].

Catalytic activities of the GenSLM-TrpBs and controls were evaluated at both room temperature and 75 °C. The elevated temperature condition was chosen to match the optimal temperature of *Pf*TrpB-0B2 and to probe the thermostability of the generated TrpBs. Residual activity from endogenous *E. coli* TrpS made it difficult to distinguish low-activity GenSLM-TrpBs from non-functional ones at room temperature. Despite this, 11 GenSLM-TrpBs showed activity clearly above

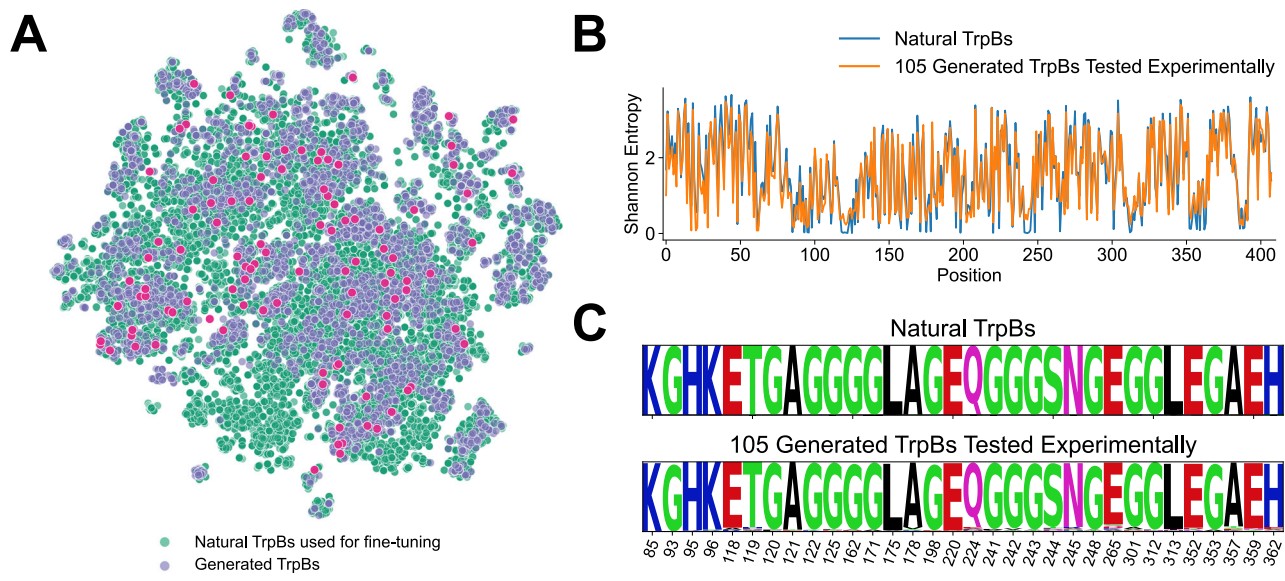

**Fig. 2 | Comparison of natural- and GenSLM-TrpBs. A** t-SNE projection showing that GenSLM-TrpBs are well distributed across the natural TrpB sequence space used for fine-tuning. **B** Sequence variability quantified by Shannon entropy and **C** sequence logo of the most conserved residues. Both analyses indicate that GenSLM-TrpBs recapitulate natural sequence patterns.

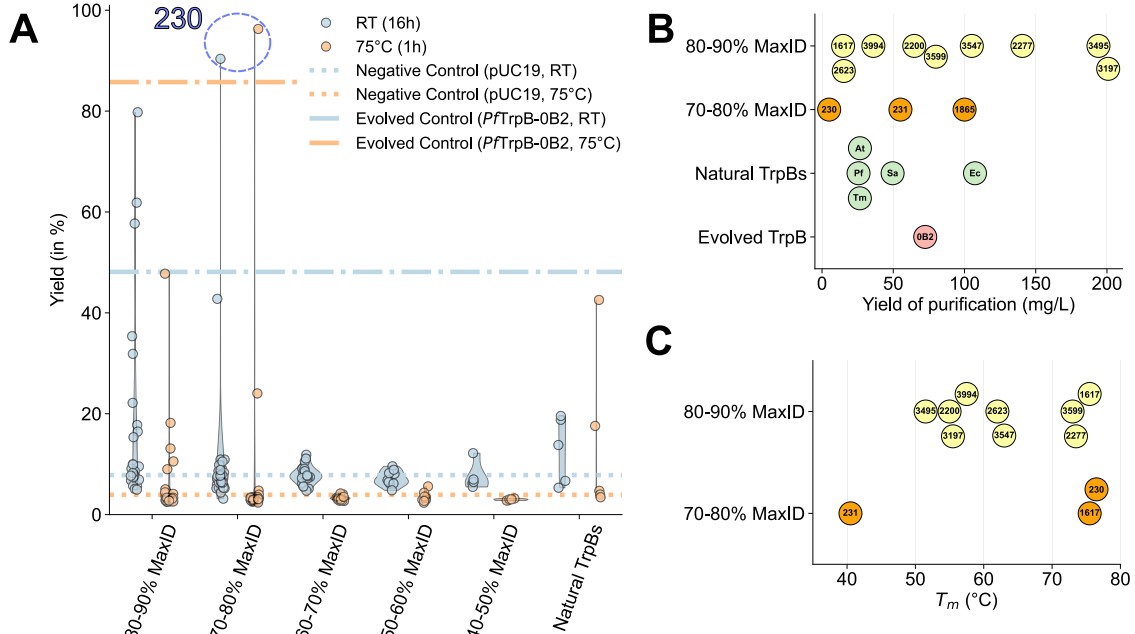

**Fig. 3 | Tryptophan production and biophysical properties of GenSLM-TrpBs. A** Truncated violin plots showing the yields of tryptophan formation catalyzed by GenSLM-TrpBs after 16 h at room temperature or 1 h at 75 °C, grouped by sequence identity to natural TrpBs and benchmarked against *Pf*TrpB-0B2. Each point represents the mean of a biological duplicate. Source data are provided as a Source Data file. **B** Expression levels of the most active GenSLM-TrpBs, reported as milligrams of isolated purified protein per liter of culture. **C** Melting temperatures ($T_m$) of the highest-performing GenSLM-TrpBs, measured by thermofluor assay. MaxID: maximum sequence identity to natural TrpBs, RT: room temperature.

background. Nine had 80–90% and two had 70–80% sequence identity to a natural TrpB. At 75 °C, seven GenSLM-TrpBs retained substantial activity, despite thermostability not being an explicit design criterion (five of these had 80–90% and two had 70–80% sequence identity to the closest natural TrpB). The results are shown in Fig. 3A and Supplementary Fig. S3.

Remarkably, several GenSLM-TrpBs exhibited levels of activity comparable to or even exceeding that of *Pf*TrpB-0B2, a variant of *Pf*TrpB specifically evolved for stand-alone function at 75 °C[26]. Given

that wild-type TrpBs require activation by their TrpA subunit, the high activity of the GenSLM-TrpBs, generated without consideration of TrpA, is noteworthy. It suggests that the model can generate sequences with properties typically acquired only through extensive laboratory evolution. Among these, **230** stands out: its total activity surpasses that of *Pf*TrpB-0B2 at both room temperature and 75 °C. This striking result establishes that a GenSLM-designed TrpB can not only rival but even surpass a laboratory-evolved benchmark enzyme, underscoring the potential of this approach.

To better characterize the most promising GenSLM-TrpBs, we purified eleven enzymes that exhibited activity at room temperature (**1617**, **2200**, **2277**, **2623**, **3197**, **3495**, **3547**, **3599**, **3994**, **230**, and **231**), as well as one (**1865**) that was active at 75 °C but inactive at room temperature. Nine of these TrpBs share 80−90% sequence identity to a natural TrpB, while the remaining three share 70−80%. Expression levels were consistently high, with an average purification yield of 84 mg/L of culture, with four enzymes exceeding 100 mg/L and two approaching 200 mg/L (Fig. 3B). We next evaluated the thermal stability of these TrpBs. Many exhibited two melting transitions, the first between 40−50 °C, that has been associated with TrpB dimer rearrangements[44]. Overall, the melting temperatures ($T_m$) were diverse: 5/12 of the tested enzymes exhibited $T_m$ above 70 °C, 6/12 were in the 50−70 °C range, and one had a $T_m$ around 40 °C (Fig. 3C and Supplementary Fig. S5).

Finally, activity assays on purified proteins conducted under varied conditions (Supplementary Fig. S4) corroborated the activity measured in 96-deep-well lysates at room temperature. However, at 75 °C, only TrpBs from thermophilic organisms (*Tm*TrpB, *Pf*TrpB, and *Pf*TrpB-0B2) retained activity, whereas the GenSLM-TrpBs lost activity. This likely reflects the lower intrinsic thermostability of the GenSLM-TrpBs, which becomes apparent after purification, in contrast to the thermophilic wild-type enzymes that maintain stability under these conditions. Purification itself is known to reduce stability through factors such as buffer exposure, removal from the intracellular milieu, and a single freeze-thaw[45,46]. Additionally, differences in heat transfer between single-vial and plate-based assay formats could further accelerate denaturation. Interestingly, while **230** was among the most active in lysate, its performance decreased following purification. In contrast, **3599**, **3994**, and **3547** exhibited superior activity in the purified format, achieving near-quantitative conversions under shorter reaction times and reduced catalyst loadings.

## GenSLM-TrpBs display broader substrate promiscuity than their natural counterparts

We hypothesized that TrpB generated via a PLM might exhibit broader substrate promiscuity compared with their counterparts that evolved naturally. This hypothesis stems from the observation that natural enzymes, like TrpB are typically highly specific, having undergone millions of years of evolutionary pressure to suppress promiscuous activities that could disrupt cellular homeostasis[10,47]. PLM-generated (or other computationally designed) enzymes may be less optimized and inherently more permissive, potentially offering greater substrate flexibility. Enzymes reconstructed through Ancestral Sequence Reconstruction (ASR), for example, have been shown to display a broader substrate range[48]. While this has often been attributed to the re-creation of a more permissive ancestral state, it remains unclear whether the observed promiscuity arises from genuine ancestral features or from biases introduced by the reconstruction method itself.

To investigate this, all 105 GenSLM-TrpBs were screened for activity on a panel of non-cognate substrates using lysates in a 96-deep-well format. While natural tryptophan synthase (TrpS) enzymes can accept substituted indoles, the activity is typically limited, and their substrate range is narrow. Reported indole substitutions accepted by the native enzymes include halogens and electron-donating substituents such as methyl, amino, methoxy, or hydroxy groups[49–52]. To challenge the generated TrpBs, we selected seven substrates that are poorly reactive with wild-type TrpBs and therefore have been targeted in prior directed evolution studies. These included 4-NO₂-, 5-NO₂-, 6-CN-, and 7-CN-indole. Naphthol is a non-indole compound that no natural TrpS has been reported to process[30]. L-threonine was also tested as an alternative electrophile, as natural TrpS enzymes display strict specificity for L-serine[29]. Finally, 5-fluoroindole was included due to its relevance in the industrial synthesis of enlicitide decanoate (Fig. 1C), although it is known to be accepted by natural TrpBs[39,51].

We compared the generated TrpBs against enzymes that were specifically evolved in the laboratory for different activities: *Pf*TrpB-0B2[26] (*Pf*TrpB stand-alone), *Tm*Triple[53] (*Tm*TrpB stand-alone), *Tm*9D8*[54] (activity on 4-CN-indole at lower temperatures), *Pf*0A9 and *Pf*2A6[28] (activity on 4-NO₂-indole and indole derivatives), *Pf*2B9[29] (activity with L-threonine), and *Tm*TyrS6/*Tm*9D8* E105G[30] (activity for phenol/naphthol to make tyrosine derivatives). Wild-type TrpBs from five species (*Ec*TrpB, *At*TrpB, *Pf*TrpB, *Tm*TrpB, and *Sa*TrpB) were included for comparison. Product identity was confirmed by comparison to authentic standards (chromatograms can be found in Supplementary Fig. S6), and reaction yields were estimated at the isosbestic point (277 nm)[31]. The results are shown in Fig. 4B, with the complete data available in Supplementary Fig. S7.

Strikingly, for every substrate tested, at least one GenSLM-TrpB exhibited measurable activity. In particular, GenSLM-TrpBs with higher sequence identity to natural TrpBs (70−80% and 80−90%) contain enzymes that showed greater promiscuity compared to the natural TrpBs, while the laboratory-evolved TrpBs consistently displayed high promiscuity (Fig. 4C). A few GenSLM-TrpBs were active on the most challenging substrates—4-NO₂-indole, naphthol, and threonine—where only **230** consistently produced UV-detectable products. Nevertheless, some additional GenSLM-TrpBs showed detectable activity by mass spectrometry, providing viable starting points for directed evolution. In contrast, a broader range of active enzymes was identified for 5-NO₂-indole, 6-CN-indole, and 7-CN-indole, with many outperforming the natural TrpBs. For 5-fluoroindole, a high background activity from endogenous *E. coli* TrpS was observed. Yet several GenSLM-designed enzymes, such as **230**, **1617**, and **3599** achieved impressive yields (99%, 97%, and 60% respectively), significantly exceeding the performance of natural TrpBs and almost reaching the quantitative yield of *Pf*TrpB-0B2, which is used industrially for this exact reaction.

Among these, **230** was particularly notable; it exhibits measurable activity across all tested substrates with yields ranging from 5% to 99% (Fig. 4D). This degree of substrate promiscuity is unprecedented among natural TrpBs. At 37 °C, **230** matched or exceeded the performance of the laboratory-evolved enzymes in reactions with 4-NO₂-indole, 5-NO₂-indole, L-threonine, and 7-CN-indole. **3599** exhibited detectable activity across the entire substrate panel, though at lower levels than **230**. **1617** was active on six of the seven tested substrates. Figure 4D compares the product yields of these two most promiscuous GenSLM-TrpBs with those of the most promiscuous natural TrpB (*At*TrpB), the most promiscuous evolved TrpB *Pf*2B9, and the industrially relevant *Pf*TrpB-0B2.

## GenSLM introduces functional improvements beyond natural sequence diversity

To better understand the origin of the properties observed in the GenSLM-TrpBs, we asked a central question: do these features merely reflect the natural sequence diversity captured by the model, or do the model introduce novel attributes beyond what is found in nature?

We previously demonstrated that GenSLM-generated sequences broadly span the natural TrpB sequence space while preserving key structural and evolutionary constraints. Notably, catalytically active GenSLM-TrpBs are evenly distributed across this space, indicating that the model sampled multiple distinct functional solutions rather than converging on a single sequence cluster (Supplementary Fig. S8). Despite this close identity with natural sequences, GenSLM-TrpBs exhibited enhanced substrate promiscuity relative to all tested natural homologs. As testing the promiscuity of all the 57,000 known natural TrpBs is impractical, we instead focused on a single representative GenSLM-TrpB, **230**, which exhibited both high catalytic activity and broad substrate range. To assess the promiscuity of **230** compared to natural TrpBs, we looked at its closest natural homolog, TrpB from *Neobacillus drentensis* (*Nd*TrpB, NCBI ID: WP_335697934.1), which shares 80.5% sequence identity (322 of 400 residues) and had not been

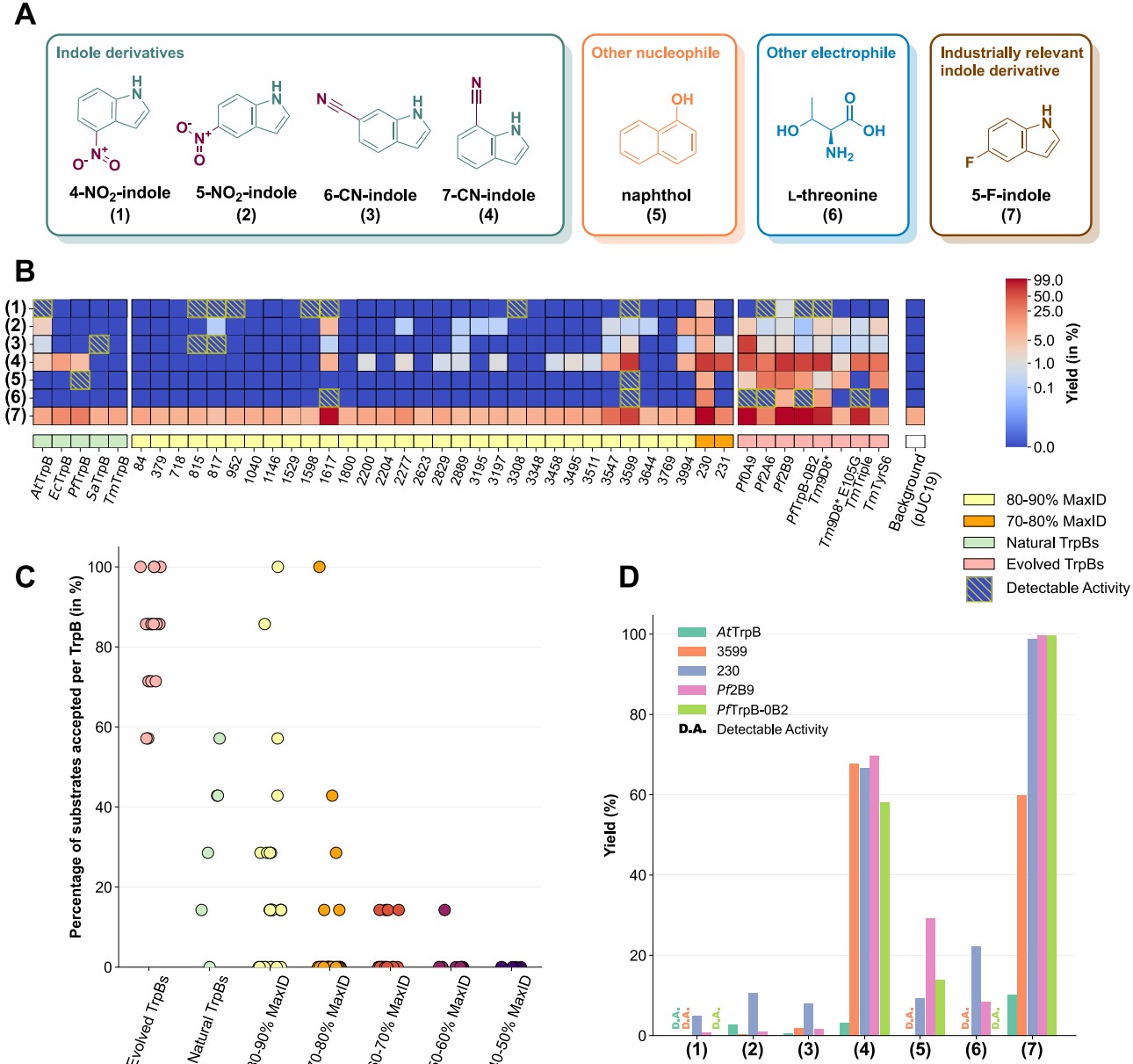

**Fig. 4 | Reaction yields of GenSLM-TrpBs with non-natural substrates.**
**A** Structures of non-canonical substrates tested. **B** Product yields estimated by UV absorbance at the isosbestic point (277 nm). Yields are displayed using a power-law normalization with $\gamma = 0.15$. Background activity corresponds to pUC19-containing *E. coli*, showing that only 5-F-indole exhibits background activity from the *E. coli* expression host. **C** Percentage of substrates accepted per TrpB, comparing GenSLM-TrpBs (clustered by sequence identity) with evolved and natural TrpBs. **D** Comparative yields of the two most promiscuous GenSLM-TrpBs compared to the most promiscuous natural enzyme (*At*TrpB), the most promiscuous evolved TrpB (*Pf*2B9), and the industrially relevant *Pf*TrpB-0B2. Detectable Activity is defined as when the product is measurable by mass spectrometry above background level (three standard deviations above pUC19-containing *E. coli*) but below UV detection on the 277 nm HPLC UV channel (see *Methods* for details). MaxID: maximum sequence identity to natural TrpBs. Source data are provided as a Source Data file.

previously characterized. Structural modeling using AlphaFold3[55] predicted identical folds, with a backbone RMSD of 0.36 Å (Fig. 5B). The active site residues are highly conserved with only one conservative substitution (V → I236) differing by a single methyl group. Figure S9 explicitly shows all mutated residues and highlights the COMM domain. All mutations appear to be located in the terminal region or within/near the loop regions. The gene encoding *Nd*TrpB was synthesized, expressed in *E. coli*, and the enzyme was purified alongside **230** using the same methods. *Nd*TrpB was expressed at a higher level (75 mg/L of culture) than **230** (5 mg/L), but they exhibited comparable thermal stability ($T_m = 76.5\,°C$ for *Nd*TrpB and $T_m = 77.5\,°C$ for

**230**). Catalytically, however, they differed substantially. Using *E. coli* lysate at room temperature, both enzymes showed high product yields (94% for *Nd*TrpB and 92% for **230**). However, at 75 °C, *Nd*TrpB activity dropped sharply to 19%, whereas **230** retained nearly full activity (94%). More strikingly, **230** exhibited significantly broader reactivity across a panel of non-canonical substrates, outperforming *Nd*TrpB in all cases. *Nd*TrpB showed detectable activity only with 7-cyanoindole and 5-fluoroindole, substrates previously known to be accepted by natural TrpBs[25], and failed to produce any product with more challenging substrates (Fig. 5A). Raw chromatograms can be found in Supplementary Fig. S10.

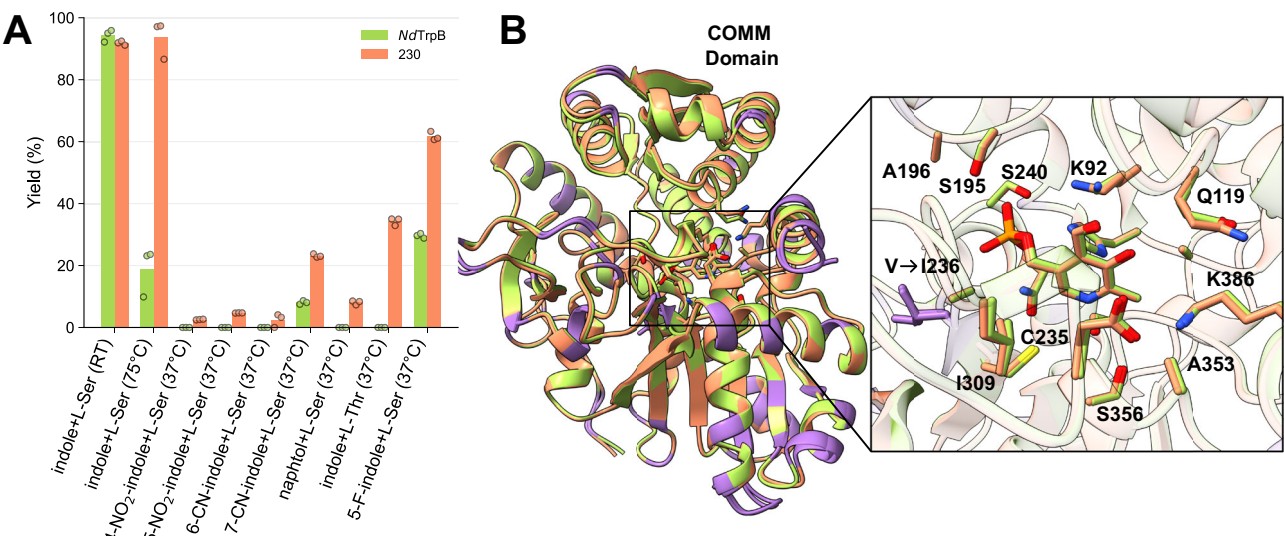

**Fig. 5 | Comparison of GenSLM-TrpB 230 and its closest natural homolog from *Neobacillus drentensis* (*Nd*TrpB). A** Product yields with the native substrate at room temperature and 75 °C, as well as with various non-canonical substrates. Bar heights represent the mean of technical triplicates (*n* = 3). Source data are provided as a Source Data file. **B** Structural alignment predicted by AlphaFold3[55] (backbone RMSD = 0.36 Å); **230** is shown in orange, *Nd*TrpB in green, and non-conserved residues are highlighted in purple. The close-up depicts the active site residues within 5 Å of PLP. Only one active site residue differs between the two structures; the mutation from **230** to *Nd*TrpB is indicated.

## Discussion

Protein engineering is undergoing a major transformation driven by advances in artificial intelligence (AI), which is reshaping how enzymes are designed and optimized. AI models trained on large biological datasets can generate enzymes with remarkable diversity and functional performance[15,56,57]. While de novo protein design enables the creation of proteins able to catalyze non-natural reactions[58,59], this approach remains limited to relatively simple reactions and is currently not applicable to highly complex systems like TrpB, whose mechanism involves two substrates, a cofactor, dynamic conformational changes, and a multistep catalytic cycle. Nevertheless, our results show that GenSLM-TrpBs were expressible, catalytically competent, stable, and broadly promiscuous. Remarkably, several GenSLM-TrpBs outperformed both natural and laboratory-evolved TrpBs on the native indole substrate, as well as on non-canonical substrates. Among the designs tested, **230** emerged as an extraordinary example. This enzyme outperforms the extensively engineered *Pf*TrpB-0B2 in tryptophan formation at both room temperature and 75 °C, while achieving comparable yield for the synthesis of 5-fluorotryptophan. Given *Pf*TrpB-0B2's industrial relevance and its long history of optimization for stand-alone activity, the discovery of a superior, PLM-generated enzyme was both surprising and exciting. Beyond its exceptional catalytic activity, **230** also displays unprecedented substrate promiscuity, catalyzing all tested non-native reactions, a property not observed in the natural TrpBs tested. Direct comparison between **230** and its closest natural homolog (*Nd*TrpB) showed that, although both enzymes display similar activity at room temperature, the natural homolog lacks high-temperature activity and broad substrate scope. This confirms that the versatility of **230** cannot be explained as simply reproducing a natural enzyme with similar properties.

The enhanced activity and substrate promiscuity of GenSLM-TrpBs cannot be attributed simply to their divergence from natural sequence space. Instead, GenSLM-TrpBs fall within the distribution of natural TrpBs, retaining catalytic activity and overall structure while diverging in terms of promiscuity. A detailed comparison between the structures of **230** and *Nd*TrpB indicates that residues in the active site are well conserved, and that most mutations are located in terminal regions or within/near loop regions. These proteins nonetheless differ by 78 amino acid mutations, making it difficult to rationalize the impact of these changes, especially given that even slight sequence changes can dramatically alter activity. For example, evolved TrpBs show dramatic changes in promiscuity, even though they carry only a few mutations relative to their closest natural TrpBs. Significant changes in promiscuity profiles have also been observed under neutral evolution, where variants are selected based on retention of activity rather than improvement. Early studies by our group and Tawfik's group demonstrated that such neutral evolution can produce enzymes with drastically altered promiscuity while bearing only a few mutations relative to the parental enzymes[60,61]. For TrpB, this was demonstrated through in vivo evolution, which generated numerous variants from the same parent enzyme, each exhibiting distinct promiscuity profiles[31].

The reasons underlying the high activity and the increased promiscuity observed in GenSLM-TrpBs are subtle and difficult to pinpoint. Multiple components of our model could influence the distribution and properties of the generated proteins, including the composition of the natural *trpB* database, the generative model and its pretraining, the fine-tuning strategy, and the filtering applied. Understanding how these factors shape protein properties is of great interest, as it could inform parameter choices for designing improved biocatalysts. We argue that the minimal filtering used here is unlikely to be responsible for the observed high activity and increased promiscuity. Filters were intentionally kept permissive to avoid introducing unwanted biases, and they do not substantially alter the overall distribution of sequences. Instead, the enhanced promiscuity likely arises from the generation process itself. GenSLM was pretrained on 110 million prokaryotic genes. It is also a model that operates at the codon level rather than the amino acid level, whose influence has not been comprehensively studied. Different fine-tuning strategies may produce distinct outcomes, and the fine-tuning database itself, composed solely of prokaryotic DNA sequences from BV-BRC, could also play an important role. A thorough, systematic dissection of these contributions, combined with experimental validation, would be highly challenging. We therefore encourage the community to explore similar generative approaches using alternative language models, fine-tuning strategies, and different protein families to determine whether enhanced promiscuity is an intrinsic property or if it is specific to our model and system.

Interestingly, a comparable effect has been observed in ASR, another data-driven approach that infers ancestral sequences from evolutionary data. Similar to what was observed in this study, ASR-generated proteins often exhibit higher promiscuity and stability than their modern counterparts, properties that have been successfully leveraged in protein engineering[48,62]. The origin of this effect remains unclear. While it has been attributed to the recovery of inherently more promiscuous ancestral enzymes, it may instead stem from the reconstruction methodology itself. For instance, a 2021 study suggested that ancestral sequences may actually reflect consensus-like sequences or yet-to-be-identified homologs, effectively bridging gaps in protein sequence space and raising the possibility that a similar mechanism might operate in our GenSLM-TrpBs[63].

These results underscore the potential of AI-driven design for making new and useful enzymes. In an optimized pipeline, the entire cycle of design, gene synthesis, and experimental testing can be completed in as little as one month, providing an exceptionally rapid route to functional enzymes. Our work shows that the GenSLM-generated enzyme library combines high activity with broad substrate scope, making it an ideal starting point for the exploration of new substrates and the evolution of new enzymatic functions while reducing experimental burden. Moreover, the fact that some generated enzymes already rival or even surpass laboratory-evolved enzymes suggests that, in some cases, it could eliminate the need for directed evolution altogether, offering a formidable acceleration for biocatalyst design, a major bottleneck limiting biocatalyst use at an industrial scale.

## Methods

### General experimental methods

All chemicals were obtained from commercial suppliers and used without further purification. Analytical liquid chromatography-mass spectrometry (LC-MS) was performed using an Agilent 1260 Infinity II LC/MSD-iQ system. PCR reactions were carried out on an Eppendorf Mastercycler X50s. The 96-well deep-well plates were shaken using an INFORS HT Multitron Shaker at 220 rpm, 80% humidity at the given temperature. Isolation of plasmids was realized using the Monarch Miniprep Kit (NEB, Ipswich, MA) according to the manufacturer's protocol. All enzymes in this study contain a C-terminal $6 \times$ His tag to enable affinity purification.

### Cloning and transformation

The pET22b(+) vector was linearized by inverse PCR using Phusion polymerase (NEB, Ipswich, MA) and primers 007 and 008 (008_Forward: CTCGAGCACCACCACCACCACCACTGAGATCCGGC; 007_Reverse: CAT ATGTATATCTCCTTCTTAAAGTTAAACAAAATTATTTC)[64] digested with DpnI (NEB, Ipswich, MA), gel-purified, and validated to ensure minimal background transformation. A total of 95 synthetic genes were synthesized by Elegen Corp. (San Carlos, CA). DNA fragments containing flanking regions (upstream: GTTTAACTTTAAGAAGGAGATATACAT; downstream: CTCGAGCACCACCATCACCACCACTGA) for Gibson assembly were received as dry residues in 96-well plates at approximately 2 µg per well. These DNA samples were dissolved in PCR-grade water to a concentration of approximately 50 ng/µL. Following NEB recommendations, 1 µL of DNA fragment was mixed with 0.5 µL of linearized pET22b(+) (200 ng/µL) and 5 µL of Gibson assembly mix in a 96-well PCR plate. The plate was sealed and incubated at 50 °C for 60 min, then placed on ice. Subsequently, 5 µL of chemically competent E. coli T7 Express cells (NEB, Ipswich, MA) were added to each well, followed by a 20 min incubation on ice and a 10 s heat shock in a 42 °C water bath. Then, 100 µL of Luria-Bertani (LB) medium were added to each well, and 10 µL of the mixture were used to inoculate 500 µL of LB containing 100 µg/mL ampicillin (LB$_{amp}$) in a 96-deep-well plate and incubated at 37 °C for 16–18 h. Cultures were passaged once by reinoculating 10 µL into fresh 500 µL LB$_{amp}$. Successful transformants were

verified via in-house sequencing (LevSeq)[65]. Additional genes (10 GenSLM-TrpBs, NdTrpB) were synthesized by Twist Bioscience (South San Francisco, CA) and processed following a similar protocol, with the exception that transformed cells were plated on LB$_{amp}$ agar instead of being inoculated into liquid media. Plasmids were isolated from selected colonies and sent for sequencing (Transnetyx, Inc., Cordova, TN). Validated plasmids were transformed into E. coli T7 Express, and glycerol stocks were prepared for long-term storage. Plasmids encoding control and previously characterized variants were retrieved from our in-house collection and transformed using the same procedure. For screening native activity, two 96-well plates were assembled containing all GenSLM-TrpBs along with control constructs. These included wild-type TrpBs (AtTrpB, PfTrpB, TmTrpB, and SaTrpB), the evolved variant PfTrpB-0B2, the empty vector pUC19 (negative control), and a sterile well. For promiscuity screening, two additional plates were assembled by supplementing the initial set with previously engineered TrpB variants: TmTriple[53], Tm9D8*[54], Pf0A9[28], Pf2A6[28], Pf2B9[29], TmTyrS6[30], and Tm9D8* E105G[30]. Control variants were included on each plate to ensure consistent comparisons across the different plates. Glycerol stocks of all transformants were prepared and archived for long-term storage and for future inoculation.

### Analytical scale analysis in 96-well plate

Glycerol stocks were used to inoculate 300 µL LB$_{amp}$ in 96-well plates, covered with a sterile, breathable film, and grown at 37 °C overnight. From the stationary-phase cultures, 50 µL were transferred into 900 µL TB containing 100 µg/mL ampicillin (TB$_{amp}$) and incubated for 2 h at 37 °C prior to induction with 50 µL of IPTG in TB$_{amp}$ (0.5 mM final). Induced cultures were incubated for 22 h at 22 °C to allow protein expression. Cells were then harvested by centrifugation ($4000 \times g$, 5 min), and the resulting pellets were either processed immediately or stored at $-20$ °C for later use.

All reactions were run using E. coli lysate. For lysate preparation in 96-well plates, cell pellets were resuspended in 500 µL of lysis buffer (100 mM KPi, pH 8.0, supplemented with 100 µM PLP, 1 mg/mL lysozyme, 2 mM Mg$^{2+}$, and DNase I) and incubated at 37 °C for 1 h with shaking at 200 rpm. Lysis was completed via three freeze-thaw cycles ($\geq 5$ min in an ethanol/dry ice bath, $> 30$ min thaw at room temperature, followed by $>5$ min in a 37 °C water bath). Cell debris was removed by centrifugation ($6000 \times g$, 15 min), and 300 µL of clarified lysate were transferred to fresh 96-deep-well plates.

For the canonical L-tryptophan synthesis assay, 10 mM L-serine were added and the volume was adjusted to reach 390 µL using 100 mM KPi buffer. Reactions were initiated by adding 10 µL of indole (400 mM in ethanol; final cosolvent concentration 2.5%). All assays were performed in biological duplicate and incubated either at room temperature for 16 h or at 75 °C for 1 h in a pre-warmed water bath. Reactions were quenched by addition of 400 µL acetonitrile (MeCN), vortexed, and centrifuged at $6000 \times g$ for 10 min. From the supernatant, 300 µL were mixed with 600 µL MeCN, clarified again by centrifugation, and 200 µL were transferred to 96-well Agilent plates for LC-MS analysis. The final 6-fold dilution in MeCN ensured the removal of salts, proteins, and particulate matter prior to LC-MS injection.

For promiscuity screening, 10 mM L-serine or L-threonine were added and the volume was adjusted to reach 340 µL using 100 mM KPi buffer. Reactions were initiated by the addition of 10 µL of indole derivatives (350 mM in EtOH or DMSO; final cosolvent concentration 2.9%). Plates were incubated for 20 h at 37 °C in a Kuhner Shaking incubator at 160 rpm. Reactions were quenched with 700 µL of a 3:1 mixture of MeCN and 1 M HCl, vortexed, and centrifuged ($6000 \times g$, 10 min). A 300 µL aliquot of the supernatant was diluted with 600 µL MeCN, enabling a final 6-fold dilution in MeCN, clarified by a second centrifugation, and 200 µL were transferred to 96-well Agilent plates for LC-MS analysis.

## LC-MS screening

The worked-up samples were transferred into assay plates, sealed, and analyzed by LC-MS using a reversed-phase Poroshell 120 EC-C18 column ($4.6 \times 50$ mm, 2.7 μm) equipped with a C18 guard column. The chromatographic method employed a solvent system of $H_2O$/MeCN with 0.1% acetic acid at a flow rate of 1 mL/min. The gradient started at 5% MeCN for 0.5 min, increased linearly to 95% MeCN over 1.5 min, held at 95% for 0.7 min, then decreased back to 5% MeCN in 0.3 min, followed by a 1-min post-run equilibration. Yields for canonical reactions were quantified by UV absorbance at 277 nm, corresponding to the isosbestic point between indole and tryptophan. For promiscuity screening, standard products were prepared from previously evolved TrpBs, and yields were similarly estimated at 277 nm, as described previously[31]. When product levels were below the UV detection limit (used to quantify yield at the 277 nm HPLC channel corresponding to the isosbestic point), products were considered detected if the ion count peak area in the mass spectrometry channel exceeded three times the standard deviation of the negative control average (strain containing pUC19), thereby ensuring rigorous exclusion of background signal.

## Protein purification

For large-scale expression, single colonies were inoculated into $LB_{amp}$ and grown overnight. The following day, $TB_{amp}$ media were inoculated at a 1:100 dilution from the overnight culture and incubated at 37 °C until the culture reached an $OD_{600}$ of 0.6–0.8. Protein expression was induced with 0.5 mM IPTG and allowed to proceed at 22 °C for 20–22 h. Cells were harvested by centrifugation ($5000 \times g$, 10 min) and stored at −20 °C until further use. Lysates were clarified by centrifugation ($>15,000 \times g$, 30 min) and loaded onto 1-mL HisTrap column using an AKTA Xpress system preequilibrated with buffer A (50 mM KPi, 200 mM NaCl, 20 mM imidazole, pH 8.0). Columns were washed with 10 column volumes (CV) of buffer A and proteins were eluted using a gradient to buffer B (50 mM KPi, 200 mM NaCl, 400 mM imidazole, pH 8.0). Eluted proteins were buffer-exchanged by dialysis into 100 mM KPi pH 8.0, aliquoted, flash-frozen in liquid nitrogen, and stored at −80 °C. Each aliquot was thawed only once and used on the same day. Protein concentrations were determined by absorbance at 280 nm using predicted extinction coefficients and molecular weights. Protein purity was evaluated by SDS-PAGE (Mini-PROTEAN TGX, 4–20%) using the Precision Plus Protein™ Kaleidoscope™ ladder, following the manufacturer's instructions.

## Analytical-scale analysis with pure protein

Proteins were thawed on ice and normalized to the specified concentration. Pyridoxal phosphate (PLP) was added at a fivefold molar excess relative to protein (PLP-only control does not catalyze the reaction). Reactions were prepared in 200 μL volumes in triplicate in Eppendorf tubes, containing 10 mM indole, 20 mM L-serine, and 5% ethanol as co-solvent. For reactions at 75 °C, tubes were incubated in a pre-warmed water bath, while those at 25 °C were incubated in a Kuhner shaker incubator at 160 rpm. Upon completion, 1 mL of MeCN was added to quench the reactions, followed by vortexing and centrifugation at maximum speed for 10 min. The clarified supernatant was then used for LC-MS analysis.

## Analytical-scale analysis with lysate

To compare the activity of **230** and *Nd*TrpB, fresh transformants were inoculated into 5 mL $LB_{amp}$ and grown overnight. The following day, 50 mL $TB_{amp}$ cultures were inoculated at a 1:100 dilution from the overnight culture and grown at 37 °C until reaching an OD of 0.6–0.8. Protein expression was induced with 0.5 mM IPTG and continued at 22 °C for 20–22 h. Cells were harvested by centrifugation ($5000 \times g$, 10 min) and stored at −20 °C. Cell pellets were resuspended in lysis buffer at an OD of 30 (100 mM KPi, pH 8.0, 100 μM PLP, DNase I, 1 mg/

mL lysozyme, 2 mM $Mg^{2+}$), incubated at 37 °C for 1 h, and lysed by sonication (2 min, 1 s on/1 s off, 35% amplitude). For each reaction, 300 μL of lysate was mixed with the electrophile (L-serine or L-threonine) and the corresponding indole derivative dissolved in DMSO or ethanol to a final cosolvent concentration of 2.5%. Reactions were performed in triplicate and processed as described in the "*Analytical scale analysis in 96-well plate*" section.

## Melting temperature determination

Melting temperatures ($T_m$) were determined using a thermofluor assay with SYPRO Orange (Invitrogen, catalog no. S6650) on a Bio-Rad CFX96 Touch Real-Time PCR system. Purified enzymes were normalized to 5 μM and supplemented with 10 μM PLP. A total volume of 150 μL of the protein-PLP mixture was combined with 10 μL SYPRO Orange, 200X stock diluted from 5000X stock in assay buffer (100 mM KPi, pH 8.0). Three 50 μL aliquots were transferred into a qPCR 96-well plate. The temperature was increased from 25 to 99 °C in 0.5 °C increments, holding each step for 30 seconds with fluorescence measurements taken at the end of each interval.

## Comparison between generated and natural TrpBs

GenSLM embeddings of both natural and generated TrpB sequences were used to create dimensionality reduction plots. Principal Component Analysis (PCA) and t-distributed Stochastic Neighbor Embedding (t-SNE) were performed using Scikit-Learn. Uniform Manifold Approximation and Projection (UMAP) visualizations were generated with the Python UMAP package, employing parameters n_neighbors=15 and min_dist=0.5, while all other settings were left at their defaults.

To assess positional variability, natural and generated sequences were aligned using MAFFT. High-gap regions were trimmed to produce a final alignment length of 408 amino acids, corresponding to the average length of natural TrpBs. Positional variability was quantified using Shannon entropy, and sequence logos were generated from this alignment using Logomaker (Bioconda).

The structures of **230** and *Nd*TrpB were predicted using AlphaFold3[55] using "PLP" as ccdCodes. The predicted Template Modeling score (pTM) and the interface pTM (ipTM) for both predicted enzymes are 0.95 and 0.98, respectively.

## Fine-tuning

The TrpB sequences were retrieved from the Bacterial and Viral Bioinformatics Resource Center (BV-BRC) database using EC classification number 4.2.1.20. Low-quality sequences were filtered out by keeping only sequences coming from complete genomes with less than 5% contamination and that contain less than 50 contigs. Since the EC class matches both the alpha and beta subunits, only features annotated as the beta subunit were included in the final dataset. Fine-tuning was performed for 10 epochs with full parameter updates, using the standard autoregressive cross-entropy objective at the codon level. No diffusion-based hierarchical modeling or reward-guided generation was used in this stage. Following fine-tuning, the model was used to generate a batch of 10,000 *trpB* gene sequences, each prompted with the methionine start codon (ATG). Sequence generation employed nucleus sampling (top_p = 0.9) in combination with top-k filtering (top_k = 50), with temperature set to 1.0. Generation was bounded to a maximum of 512 codon tokens, with early termination if a designated stop token was encountered. Sequences were generated in batches of 512.

## Filtering pipeline

A schematic depiction of the filters is shown in Supplementary Fig. S1. It is important to note that the order of filters is determined to maximize the novelty and functionality of the filtered sequences. The fine-tuned GenSLM generates *trpB* codon sequences in an autoregressive

manner, conditioned on an initial token provided as a prompt. To streamline the downstream filtering process, all generated nucleotide sequences were translated into their corresponding amino acid sequences.

**Start codon filter.** We first filtered sequences to retain only those that begin with the canonical start token as the **ATG** codon. This step ensures translational initiation compatibility and reflects biological realism. All *trpB*-prompted sequences passed this filter.

**Length filter.** Next, we enforced a length constraint based on the natural *trpB* sequence distribution (references). Sequences were accepted if their length $L$ satisfied:

$$\mathcal{L} - 2\sigma \leq L \leq \mathcal{L} + 2\sigma, \tag{1}$$

where $\mathcal{L} = 363.55$ and $\sigma = 57.91$—statistics obtained from the references, result in a valid range of 306–421 amino acids. This filter ensures the resulting sequences remain within a functional length range typical of the protein family. Among *trpB*-prompted sequences, 89% passed the length filter.

**Structural integrity filter.** We then assessed structural integrity using ESMFold[43]. Only sequences with predicted Local Distance Difference Test (pLDDT) scores ≥80% were retained, indicating high folding confidence of the protein backbone. This filter removed sequences unlikely to adopt stable structures. Of the remaining length-filtered sequences, 93% passed the structural confidence threshold.

The autoregressive nature of GenSLM makes the initial token critical, as it influences downstream sequence generation. Additionally, maintaining sequence lengths similar to natural variants is essential, as protein function and folding are often length-dependent within a given family. Applying these filters to a batch of 5000 generated TrpBs resulted in pass rates of 95% for length and 90% for structural confidence (Fig. S1B).

**Optional Filter 1: stability filter.** We also evaluated Rosetta energy scores[66] as a proxy for thermodynamic stability. Although this metric can be used to remove unstable sequences, we did not apply a hard threshold in this round of selection. First, folding confidence (pLDDT) and Rosetta energy are correlated, so unstable sequences are already partly filtered. Second, strictly filtering by stability may reduce the novelty of the generated proteins and bias selection toward variants that function only under specific conditions, thus limiting potential discovery.

**Sequence novelty via Max Identity.** To assess novelty, we computed the maximum sequence identity (MaxID) of each filtered sequence relative to a reference set of natural TrpBs, following REF[19]. Figure S1C shows the distribution of MaxID values, where it peaks near 100% identity, reflecting the high fidelity of the generative model and the stringency of the initial filters.

**Partitioning by sequence identity.** To systematically explore sequence novelty, we partitioned the filtered sequences into six identity bins based on max ID: (40–50%), (50–60%), (60–70%), (70–80%), (80–90%), and (90–100%). This partitioning enables balanced sampling of sequences across different similarity levels while retaining plausible function. Figure S1C shows the population size within each bin.

**Ranking by alignment score.** Within each bin, we prioritized sequences using a custom **Alignment Score**, defined as a weighted sum of global and local pairwise alignment scores to all reference sequences. Grid search identified optimal weights: 3 for global and 8

for local alignment. Scoring used +1 for residue matches, −1 for mismatches, − 0.5 for gap openings, and − 0.1 for gap extensions. Sequences were ranked accordingly, and the top candidates from each bin were selected.

**Optional Filter 2: active site conservation.** To preserve catalytic functionality, we aligned all candidates to *Pf*TrpB (PDB: 5dw0) using multiple sequence alignment and retained only sequences with a conserved lysine at position 82, the known catalytic residue.

**Optional Filter 3: sequence diversity.** To ensure diversity among final candidates, we clustered sequences using BLAST at a 70% sequence similarity threshold and selected only one representative per cluster. This step promotes coverage of distinct sequence families and reduces redundancy.

**Final selection.** All sequences with >90% identity to any natural variant were excluded. From the remaining identity bins, we selected 30 sequences with 80–90% MaxID, 40 with 70–80%, 20 with 60–70%, 10 with 50–60%, and 5 with 40–50%, totaling 105 sequences. All sequences were codon-optimized for *E. coli* expression. The ordered DNA sequences are provided in Supplementary Data 1.

### Reporting summary

Further information on research design is available in the Nature Portfolio Reporting Summary linked to this article.

## Data availability

The datasets generated and analyzed during this study, including the reference TrpB sequences, the fine-tuning dataset, and the final selected candidate sequences, are publicly available in the GitHub repository at https://github.com/AI-ProteinDesign/GenSLM-TrpB/. The DNA sequences selected for experimental validation are provided in the Supplementary Information, along with their sources. These data constitute the minimum dataset required to interpret, verify, and extend the findings of this study. Source data are provided with this paper.

## Code availability

The code used to implement the computational filtering pipeline, the pretrained GenSLM model, and the fine-tuned GenSLM checkpoints is publicly available in the same GitHub repository at https://github.com/AI-ProteinDesign/GenSLM-TrpB/ under the MIT license and can be used to reproduce the analyses reported in this study.

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

## Acknowledgements
This material is based upon work supported by the U.S. Department of Energy, Office of Science, Office of Basic Energy Sciences, under Award Number DE-SC0022218. This report was prepared as an account of work sponsored by an agency of the United States Government. Neither the United States Government nor any agency thereof, nor any of their employees, makes any warranty, express or implied, or assumes any legal liability or responsibility for the accuracy, completeness, or usefulness of any information, apparatus, product, or process disclosed, or represents that its use would not infringe privately owned rights. Reference herein to any specific commercial product, process, or service by trade name, trademark, manufacturer, or otherwise does not necessarily constitute or imply its endorsement, recommendation, or favoring by the United States Government or any agency thereof. The views and opinions of authors expressed herein do not necessarily state or reflect those of the United States Government or any agency thereof. T.L. gratefully acknowledges financial support for this research by the Fulbright Program, which is sponsored by the U.S. Department of State and the Franco-American Commission - Fulbright France. Its contents are solely the responsibility of the author and do not necessarily represent the official views of the Fulbright Program, the Government of the United States, or the Franco-American Commission. A.A. is supported by the Bren Endowed Chair and the Schmidt AI2050 Senior Fellowship. A.A. and A.T. further acknowledge support from the Eric & Wendy Schmidt Fund for Strategic Innovation (Award G-23-65858). A.T. also acknowledges funding from the U.S. Department of Energy (Award SC00022218) and from the Margot and Tom Pritzker Foundation through a subaward from the University of Chicago (Award ORS000086). The authors thank Sabine Brinkmann-Chen for critical reading of the manuscript. The authors also thank Kyle Hippe for helping with extracting and filtering the TrpB sequences and Ariane Mora for checking the integrity of the sequences.

## Author contributions
T.L. conceived the experimental study and performed the wet-lab experiments, AF3 modeling, and biostatistical analysis, wrote the manuscript, and combined the editing. A.T. was in charge of the machine learning, including model fine-tuning, data collection and filtering pipeline, and helped with editing. G.D. contributed to model fine-tuning and data collection. J.Y. provided general advice, performed quality control and helped with editing. V.B. helped in the design of the filtering pipeline. S.K. and M.H. coordinated DNA synthesis. A.R. provided resources, supervision, and funding. A.A. contributed resources, manuscript editing, supervision, and funding. F.H.A. oversaw the project, provided resources and funding, and contributed to manuscript editing and supervision.

## Competing interests
The authors declare no competing interests.
