## [Transparent Peer Review file · Nature Communications]

Sequence-Based Generative AI Design of Versatile Tryptophan Synthases

Corresponding Author: Professor Frances Arnold

Version 0:

Reviewer comments:

Reviewer #1

(Remarks to the Author)

In this study, Lambert and coworkers reports the fine-tuned GenSLM protein language model along with a series of filters to design novel tryptophan synthases (TrpBs), some of which exhibit high stability and unique catalytic properties. The key innovation of this work lies in the fine-tuning methodology of the PLM and the filtering pipeline, which significantly improve both the efficiency of TrpBs design and the catalytic performance of the enzymes. This manuscript is well-executed and the data is solid. This reviewer recommends its publication in Nature Communications after addressing the following points.

Major comments:

1. The authors found that TrpB 230 outperforms other enzymes in terms of catalytic activity and substrate promiscuity. While the structural comparison between TrpB 230 and NdTrpB is provided, I suggest a deeper analysis of the structural differences in the catalytic pockets of these two enzymes, particularly focusing on the potential reasons behind the broader substrate promiscuity observed in TrpB 230. This additional analysis could provide more insights into the structural basis for the observed functional differences.
2. The authors prepared two datasets for TrpBs: one with >57,000 sequences and another with 22,800 sequences. Typically, for fine-tuning tasks, it is standard practice to use as much data as available. I recommend that the authors explain why the smaller dataset (22,800 sequences) was chosen for fine-tuning, rather than utilizing a larger portion (e.g., 90%) of the 57,000 sequences.
3. The authors tested 105 TrpBs, of which 12 exhibited catalytic activity. I recommend that the authors compare the sequences of active and inactive enzymes to identify the preferential differences. For instance, after aligning these sequences, select and focus on comparing highly conserved residues or those in the catalytic pocket. This comparison could offer valuable insights for the development of new filters, potentially improving the process for selecting more active enzymes in future studies.

Minor comment:

1. In Figures 3A and 4B, the SeqID values described are somewhat confusing. For example, the authors did not test TrpBs with 90% sequence identity, yet the x-axis labels in the figure 3A show 90% SeqID. Should this be revised to "90-80% SeqID" instead?

(Remarks on code availability)

I have browsed the uploaded code files and read the README file, but I did not attempt to install and run the code. Based on my experience, the code files appear to be reproducible, and the README file provides detailed instructions. I did not find any obvious problems.

Reviewer #2

(Remarks to the Author)

In this work, Lambert and coworkers apply a sequence-based generative AI protocol for generating multiple tryptophan synthase (beta subunit) sequences. Using the approach, they generate multiple sequences, among which 105 were tested experimentally and compared to natural and laboratory-evolved TrpBs. One sequence (labelled as 230) turned out to have interesting properties, such as substrate promiscuity and activity at high temperatures, but at the expense of rather low expression levels. This sequence is then compared to the closest natural homologue corresponding to NdTrpB. The protocol is based on the 25M parameter GenSLM, which was fine-tuned on a curated dataset of trpB DNA sequences. The authors mention that they use a filtering scheme for prioritizing sequences that are active and distinct from natural enzymes,

however, this filtering scheme does not contain any activity filter to discriminate between active and inactive sequences (the only one included is the analysis of whether the catalytic lysine is contained or not in the sequence). Therefore, the filtering does not consider any “catalytic” information, apart from the presence of the catalytic lysine, which by itself does not guarantee tryptophan synthase activity. Related to this point, the language model generates sequences, which could potentially exhibit the desired properties for then undergoing further evolution, but it could very likely be that 0 sequences from the sequence pool have the desired properties as the training of the language model was performed considering only DNA sequences, i.e. not including any catalytic feature or information.

As shown in Figure 4C, among all sequences generated, only three of them accept a similar number of substrates as some laboratory-evolved TrpB (pink dots in the plot). Therefore, the level of promiscuity of the GenSLM-TrpB sequences is rather low (although the authors mention that “GenSLM-TrpBs were expressible, catalytically competent, stable, and broadly promiscuous”). In fact, this rather low promiscuity can also be seen in Figure 4B, which shows that the part containing more colored (i.e. higher yields and higher acceptance of substrates) corresponds to the right side where the evolved TrpBs are located. The concept of “detectable activity” marked with a yellow dashed box should also be better quantified.

Another interesting point to include is the sequence identity between the most promising sequences between them, and how they differ in terms of mutations. This should also be done for the 230/NdTrpB comparison: more details should be included on the substitution differences and where they are located within the key regions of the enzyme, namely active site, COMM domain, TrpA-TrpB interface region.

Along the years many groups have worked on the design of stand-alone tryptophan synthases, including continuous evolution strategies, ancestral sequence reconstruction, and rational enzyme design approaches. All these studies should be included and properly referenced in the introduction.

(Remarks on code availability)

Reviewer #3

(Remarks to the Author)

Lambert et al. report on the engineering of the β -subunit of tryptophan synthases (TrpB) with a fine-tuned version of the protein language model GenSLM, the primary result being the generation of functional variants with broader substrate scope than native TrpBs, and thermostability and activity similar to and sometimes exceeding that of previously evolved TrpBs. Previous efforts have utilized language models to optimize enzymes, but the approach taken here is unique in that it works at the level of codons instead of amino acids, although it is not apparent to this reviewer how that may have impacted sequence design or experimental outcomes. The generation of variants with improved function to those that required significant evolution is notable, but the similarity to native TrpB sequences does seem to reduce some of the conclusions that the approach here is free of biological constraints given the backbone of the best design is within 1 Å rmsd and has 80-90% sequence identity to a native TrpB. Regardless, the work appears well executed and folks in ML and protein engineering will I'm sure want to read about it.

The scientists show that their best variant is quite similar to an existing TrpB at the fold level (RMSD 0.87 Å), but do not mention any specific amino acid substitutions or structural analysis of the active site. I understand the hesitation to overinterpret models, especially at the sidechain level, but I think it rounds out the story, especially given that the side chains of this active site are involved in catalysis, it seems worth mentioning them. Along these lines, it would be worth predicting the structures of the ordered designs with AlphaFold3. The online server I believe includes PLP as an available ligand, and predicting the structure of the PLP-bound form I imagine could be very informative, and would be certainly interesting to see if it happens to differentiate at all between active and inactive designs. Even better would be to predict the PLP complexes with your various substrates to better understand the structural basis of the substrate promiscuity in variant 230, which is also possible with AlphaFold3, although not the online server version so it would take some effort to set up (although I think worthwhile).

Is it clear for trpB variants that do not go to completion if they inactivate after some amount of turnovers or if they simply are very slow?

What are the sequence differences that enable activity in only three of the 80% seqID designs and not all the others? The low number of hits at that sequence ID could provide insights. The same question I think is relevant for the 90% seq ID hits (using the seqID terms as described in Figure S3).

Some results of the raw data from the experimental effort would be valuable to present for transparency, even if for only the best variants. For example, an LC-MS of the enzyme reaction products or similar showing raw data corresponding to product formation. Some raw data may also help to understand the gap between ‘background’ reactivity and that which is determined ‘detectable’.

Are there any conclusions to draw about using GenSLM over any other PLM? Do you find similar sequences with other models? Is fine-tuning necessary to generate these variants? These questions may take work to answer, but it's at least worth acknowledging this at some point in the discussion. And lastly, as mentioned before, it doesn't seem like ending the story on being free from biological constraints makes much sense. The model is fine-tuned on TrpBs and anything less than

80% identical to a native enzyme didn't work. It seems better to highlight that PLMs reduce the time spent at the bench evolving enzymes and let you get on to the next thing.

(Remarks on code availability)

Reviewer #4

(Remarks to the Author)

(Remarks on code availability)

Version 1:

Reviewer comments:

Reviewer #1

(Remarks to the Author)

The authors have performed sufficient experiments and calculations in response to my comments, so I recommends its publication in Nature Communications.

(Remarks on code availability)

Reviewer #2

(Remarks to the Author)

In this new revised version of the manuscript, the authors have successfully addressed all our original concerns and comments. We believe the current version of the paper can now be published in Nature Communications as it is.

(Remarks on code availability)

Reviewer #3

(Remarks to the Author)

My comments were satisfactorily addressed and I support publication of the work.

(Remarks on code availability)

My comments were satisfactorily addressed and I support publication of the work.

Reviewer #4

(Remarks to the Author)

(Remarks on code availability)

We thank the reviewers for their thoughtful and constructive feedback, which has been invaluable in improving the clarity and overall quality of our manuscript. We have carefully revised the text to address all comments, and we believe these changes have significantly strengthened the work. Below, we provide a detailed, point-by-point response to each reviewer's remark. Each comment has been numbered for clarity and is followed by the corresponding response.

Reviewer #1 (Remarks to the Author):

In this study, Lambert and coworkers report the fine-tuned GenSLM protein language model along with a series of filters to design novel tryptophan synthases (TrpBs), some of which exhibit high stability and unique catalytic properties. The key innovation of this work lies in the fine-tuning methodology of the PLM and the filtering pipeline, which significantly improve both the efficiency of TrpBs design and the catalytic performance of the enzymes. This manuscript is well-executed and the data is solid. This reviewer recommends its publication in Nature Communications after addressing the following points.

Major comments:

1. The authors found that TrpB 230 outperforms other enzymes in terms of catalytic activity and substrate promiscuity. While the structural comparison between TrpB 230 and NdTrpB is provided, I suggest a deeper analysis of the structural differences in the catalytic pockets of these two enzymes, particularly focusing on the potential reasons behind the broader substrate promiscuity observed in TrpB 230. This additional analysis could provide more insights into the structural basis for the observed functional differences.

2. The authors prepared two datasets for TrpBs: one with >57,000 sequences and another with 22,800 sequences. Typically, for fine-tuning tasks, it is standard practice to use as much data as available. I recommend that the authors explain why the smaller dataset (22,800 sequences) was chosen for fine-tuning, rather than utilizing a larger portion (e.g., 90%) of the 57,000 sequences.

3. The authors tested 105 TrpBs, of which 12 exhibited catalytic activity. I recommend that the authors compare the sequences of active and inactive enzymes to identify the preferential differences. For instance, after aligning these sequences, select and focus on comparing highly conserved residues or those in the catalytic pocket. This comparison could offer valuable insights for the development of new filters, potentially improving the process for selecting more active enzymes in future studies.

Minor comment:

4. In Figures 3A and 4B, the SeqID values described are somewhat confusing. For example, the authors did not test TrpBs with 90% sequence identity, yet the x-axis labels in the figure 3A show 90% SeqID. Should this be revised to "90-80% SeqID" instead?

Reviewer #1 (Remarks on code availability):

I have browsed the uploaded code files and read the README file, but I did not attempt to install and run the code. Based on my experience, the code files appear to be

reproducible, and the README file provides detailed instructions. I did not find any obvious problems.

Response to Reviewer #1:

1. We agree with the reviewer's insightful comment, which was also raised by other reviewers. To address this point, we generated new structural models of TrpB 230 and *Nd*TrpB using AlphaFold3, which allows inclusion of the PLP cofactor in the predictions. Figure 5 has been updated to highlight the residues in the active site, and a new Figure S9 has been added to the Supplementary Information to provide additional visualizations and a focus on the mutations in the COMM domain. We also expanded the discussion in the main text to address these mutations, emphasizing the complexity of rationalizing their effects on enzyme activity and substrate promiscuity.

2. We appreciate the reviewer's comment. The dataset used for fine-tuning was not a smaller one but consisted of 30,000 unique DNA sequences from the BV-BRC database, corresponding to approximately 22,800 unique amino acid sequences. Fine-tuning was performed at the codon (nucleotide) level, which required a dataset containing DNA information, data not available in UniProt. The larger dataset of 57,000 sequences, comprising the BV-BRC-translated sequences and additional UniProt entries for which no corresponding DNA data are available, was used exclusively at the amino acid level for post-generation filtering and evaluation. This approach enabled robust codon-level fine-tuning while ensuring unbiased and rigorous evaluation using a broader amino acid reference set.

3. We appreciate this important comment and agree that comparing active and inactive enzymes could provide valuable insights. However, catalytic and conserved residues are largely maintained across our generated proteins, making sequence-level differences subtle and very difficult to interpret. We attempted to correlate the measured activities with physics-informed models but found no clear relationships. This is probably because loss of activity likely arises from multiple intertwined factors such as expression, folding, solubility, stability, interaction with other proteins or molecules within the expression strain, rather than specific sequence variations. Moreover, in our case sequence comparisons seem to be biased, as active GenSLM-TrpBs share higher identity with natural TrpBs compared to inactive ones. We have expanded the Discussion to include additional points addressing this aspect. The most informative analysis is presented in Figure 2, which shows residue conservation and positional variability across all tested enzymes.

Regarding the filtering strategy, we expanded the discussion in the revised manuscript and emphasized that only minimal, widely validated filters were applied to avoid

introducing bias. We therefore consider it unlikely that the observed high activity and promiscuity result from the filtering procedure.

4. All labels in the figures have been corrected to reflect the proper sequence identity values, and “SeqID” has been replaced with “MaxID” for consistency with the text.

Reviewer #2 (Remarks to the Author):

In this work, Lambert and coworkers apply a sequence-based generative AI protocol for generating multiple tryptophan synthase (beta subunit) sequences. Using the approach, they generate multiple sequences, among which 105 were tested experimentally and compared to natural and laboratory-evolved TrpBs. One sequence (labelled as 230) turned out to have interesting properties, such as substrate promiscuity and activity at high temperatures, but at the expense of rather low expression levels. This sequence is then compared to the closest natural homologue corresponding to NdTrpB. The protocol is based on the 25M parameter GenSLM, which was fine-tuned on a curated dataset of trpB DNA sequences.

1. The authors mention that they use a filtering scheme for prioritizing sequences that are active and distinct from natural enzymes, however, this filtering scheme does not contain any activity filter to discriminate between active and inactive sequences (the only one included is the analysis of whether the catalytic lysine is contained or not in the sequence). Therefore, the filtering does not consider any “catalytic” information, apart from the presence of the catalytic lysine, which by itself does not guarantee tryptophan synthase activity. Related to this point, the language model generates sequences, which could potentially exhibit the desired properties for then undergoing further evolution, but it could very likely be that 0 sequences from the sequence pool have the desired properties as the training of the language model was performed considering only DNA sequences, i.e. not including any catalytic feature or information.

2. As shown in Figure 4C, among all sequences generated, only three of them accept a similar number of substrates as some laboratory-evolved TrpB (pink dots in the plot). Therefore, the level of promiscuity of the GenSLM-TrpB sequences is rather low (although the authors mention that “GenSLM-TrpBs were expressible, catalytically competent, stable, and broadly promiscuous”). In fact, this rather low promiscuity can also be seen in Figure 4B, which shows that the part containing more colored (i.e. higher yields and higher acceptance of substrates) corresponds to the right side where the evolved TrpBs are located.

3. The concept of “detectable activity” marked with a yellow dashed box should also be better quantified.

4. Another interesting point to include is the sequence identity between the most promising sequences between them, and how they differ in terms of mutations. This should also be done for the 230/NdTrpB comparison: more details should be included on the substitution differences and where they are located within the key regions of the enzyme, namely active site, COMM domain, TrpA-TrpB interface region.

5. Along the years many groups have worked on the design of stand-alone tryptophan synthases, including continuous evolution strategies, ancestral sequence

reconstruction, and rational enzyme design approaches. All these studies should be included and properly referenced in the introduction.

Response to Reviewer #2:

1. We fully agree with this remark and have revised the description of our filtering strategy in Section 2.1 and the Discussion. Our filters were intentionally kept minimal and limited to experimentally validated criteria that are already used in the community, to avoid introducing potential biases from AI- or physics-based filters that have not been shown to correlate with experimental outcomes. As detailed in the revised Discussion, we describe the impact of our filtering scheme on the results and conclude that the observed activity and substrate promiscuity arise primarily from the sequence generation step, rather than from the filtering process, which only minimally alters the distribution of generated sequences.

2. We understand this comment and note that our manuscript does not claim that the generated TrpBs match the promiscuity of laboratory-evolved TrpBs. However, we argue that the generated sequences have a clear increase in promiscuity compared to natural TrpBs, which represents a meaningful improvement for enzyme screening, especially in contexts where evolved variants are not available. As stated in the manuscript: “GenSLM-TrpBs [...] contain enzymes that showed greater promiscuity compared to the natural TrpBs, while the laboratory-evolved TrpBs consistently displayed high promiscuity” “This degree of substrate promiscuity is unprecedented among natural TrpBs”. The apparent confusion may stem from TrpB 230, which was compared directly to evolved enzymes because its promiscuity reaches comparable levels: “At 37 °C, 230 matched or exceeded the performance of the laboratory-evolved enzymes in reactions with 4-NO₂-indole, 5-NO₂-indole, l-threonine, and 7-CN-indole.”

3. We agree with this comment and have clarified the concept of “detectable activity” in the figure legend and Methods. The Figure 4 legend now reads: “Detectable Activity is defined as when the product is measurable by mass spectrometry above background level (three standard deviation above pUC19-containing *E. coli*) but below UV detection on the 277 nm HPLC UV channel (see Methods for details)”

4. This point has been addressed in **Response to Reviewer #1, section 1.**

5. We thank the reviewer for this comment and have updated our introduction accordingly.

Reviewer #3 (Remarks to the Author):

Lambert et al. report on the engineering of the β -subunit of tryptophan synthases (TrpB) with a fine-tuned version of the protein language model GenSLM, the primary result being the generation of functional variants with broader substrate scope than native TrpBs, and thermostability and activity similar to and sometimes exceeding that of previously evolved TrpBs.

1. Previous efforts have utilized language models to optimize enzymes, but the approach taken here is unique in that it works at the level of codons instead of amino acids, although it is not apparent to this reviewer how that may have impacted sequence design or experimental outcomes.

2. The generation of variants with improved function to those that required significant evolution is notable, but the similarity to native TrpB sequences does seem to reduce some of the conclusions that the approach here is free of biological constraints given the backbone of the best design is within 1 Å rmsd and has 80-90% sequence identity to a native TrpB. Regardless, the work appears well executed and folks in ML and protein engineering will I'm sure want to read about it.

3. The scientists show that their best variant is quite similar to an existing TrpB at the fold level (RMSD 0.87 Å), but do not mention any specific amino acid substitutions or structural analysis of the active site. I understand the hesitation to overinterpret models, especially at the sidechain level, but I think it rounds out the story, especially given that the side chains of this active site are involved in catalysis, it seems worth mentioning them.

4. Along these lines, it would be worth predicting the structures of the ordered designs with AlphaFold3. The online server I believe includes PLP as an available ligand, and predicting the structure of the PLP-bound form I imagine could be very informative, and would be certainly interesting to see if it happens to differentiate at all between active and inactive designs. Even better would be to predict the PLP complexes with your various substrates to better understand the structural basis of the substrate promiscuity in variant 230, which is also possible with AlphaFold3, although not the online server version so it would take some effort to set up (although I think worthwhile).

5. Is it clear for trpB variants that do not go to completion if they inactivate after some amount of turnovers or if they simply are very slow?

6. What are the sequence differences that enable activity in only three of the 80% seqID designs and not all the others? The low number of hits at that sequence ID could provide insights. The same question I think is relevant for the 90% seq ID hits (using the seqID terms as described in Figure S3).

7. Some results of the raw data from the experimental effort would be valuable to present for transparency, even if for only the best variants. For example, an LC-MS of the enzyme reaction products or similar showing raw data corresponding to product formation. Some raw data may also help to understand the gap between ‘background’ reactivity and that which is determined ‘detectable’.

8. Are there any conclusions to draw about using GenSLM over any other PLM? Do you find similar sequences with other models? Is fine-tuning necessary to generate these variants? These questions may take work to answer, but it’s at least worth acknowledging this at some point in the discussion.

9. And lastly, as mentioned before, it doesn’t seem like ending the story on being free from biological constraints makes much sense. The model is fine-tuned on TrpBs and anything less than 80% identical to a native enzyme didn’t work. It seems better to highlight that PLMs reduce the time spent at the bench evolving enzymes and let you get on to the next thing.

Response to Reviewer #3:

1. and 8. We thank the reviewer for this insightful comment. We have expanded the Discussion to address these points. Specifically, we cannot draw conclusions about the performance of GenSLM relative to other protein language models, as no direct comparison exists. Design choices within each model can strongly influence the outputs, making such comparisons difficult. Similarly, defining “similar” sequences is challenging, as even a small number of mutations can dramatically affect experimental performance. Fine-tuning was essential in our case, as the model could not be conditioned for TrpB generation without it. In the Discussion, we now encourage the community to explore other models, fine-tuning strategies, and enzyme families to determine whether the main observation of our manuscript, i.e. the increase in promiscuity, is generalizable across AI generative approaches, as has been suggested in other computational strategies such as Ancestral Sequence Reconstruction (ASR). We also ensured that we make no claim that sequence-based models could be a better choice, as this has not been demonstrated and is not the purpose of the present study.

2. and 9. We thank the reviewer for these remarks and agree that emphasizing “freedom from biological constraints” is misleading and was not our intent. We have revised the manuscript to remove this phrasing and instead focus the Discussion on the points raised by other reviewer comments.

3. This point has been addressed in **Response to Reviewer #1, section 1.**

4. We appreciate the reviewer's suggestion and agree that this is an interesting idea. We had already performed docking studies using both AlphaFold3 and physics-based docking but did not observe any clear trends distinguishing active from inactive variants, for reasons that have been discussed in **Response to Reviewer #1, section 3**. Based on the reviewer recommendation, we predicted protein–ligand complexes using AlphaFold3 with different substrates for comparing 230 and *NdTrpB*, but the resulting structures were nearly identical and highly superimposable, regardless of the ligand. This is consistent with AlphaFold3's tendency to overfit and position different ligands similarly. The differences in promiscuity likely arise from subtle variations in the conformational ensemble, which are not captured by these approaches. More computationally demanding methods would be required to explore this mechanism in detail, but such analyses are beyond the scope of the present study and may not necessarily yield more informative insights.

5. We appreciate this insightful question that is scientifically very interesting. However, we believe it falls outside the scope of the present study.

6. We thank the reviewer for this insightful comment. Indeed, understanding why only a few variants are active at 70–80% or 80–90% sequence identity would be very interesting. However, rationalizing the effects of individual mutations across these sequences is extremely challenging, as the impact of single amino acid changes on activity is highly context-dependent (see our **Response to Reviewer #1, section 3**). Instead, we focused our analysis on the comparison between TrpB 230 and *NdTrpB*, for which we cannot even explain the effects of specific substitutions. We have expanded the Discussion to acknowledge that it is truly complicated to comprehensively rationalize the observed effects of these mutations.

7. We thank the reviewer for this valuable comment, as transparency in our data is crucial. We have added all chromatogram traces from the evolved control, which serve as a reference for retention times (new Figure S6). In addition, we included traces from GenSLM-TrpB 230 and *NdTrpB*, to strengthen our claims (new Figure S10). The concept of detectable activity has been addressed in **Response to Reviewer #2, section 3**, and we have added a heatmap showing background activity, highlighting the background activity for 5-F-indole due to the presence of TrpS in the expression strain that was only mentioned in the text and absent in the figure. We greatly appreciate the reviewer's suggestion, as these additions improve clarity and transparency in the manuscript.